# Natural Products Extracted from Fungal Species as New Potential Anti-Cancer Drugs: A Structure-Based Drug Repurposing Approach Targeting HDAC7

**DOI:** 10.3390/molecules25235524

**Published:** 2020-11-25

**Authors:** Annalisa Maruca, Roberta Rocca, Raffaella Catalano, Francesco Mesiti, Giosuè Costa, Delia Lanzillotta, Alessandro Salatino, Francesco Ortuso, Francesco Trapasso, Stefano Alcaro, Anna Artese

**Affiliations:** 1Dipartimento di Scienze della Salute, Università “Magna Græcia” di Catanzaro, Campus “S. Venuta”, Viale Europa, 88100 Catanzaro, Italy; maruca@unicz.it (A.M.); catalano@unicz.it (R.C.); francesco.mesiti@studenti.unicz.it (F.M.); gcosta@unicz.it (G.C.); ortuso@unicz.it (F.O.); artese@unicz.it (A.A.); 2Net4Science Academic Spin-Off, Università “Magna Græcia” di Catanzaro, Campus “S. Venuta”, Viale Europa, 88100 Catanzaro, Italy; rocca@unicz.it; 3Dipartimento di Medicina Sperimentale e Clinica, Università “Magna Græcia” di Catanzaro, Campus “S. Venuta”, Viale Europa, 88100 Catanzaro, Italy; delialanzillotta@unicz.it (D.L.); salatino@unicz.it (A.S.); trapasso@unicz.it (F.T.)

**Keywords:** mushrooms, HDAC7, cancer, repositioning, structure-based virtual screening, molecular dynamics

## Abstract

Mushrooms can be considered a valuable source of natural bioactive compounds with potential polypharmacological effects due to their proven antimicrobial, antiviral, antitumor, and antioxidant activities. In order to identify new potential anticancer compounds, an *in-house* chemical database of molecules extracted from both edible and non-edible fungal species was employed in a virtual screening against the isoform 7 of the Histone deacetylase (HDAC). This target is known to be implicated in different cancer processes, and in particular in both breast and ovarian tumors. In this work, we proposed the ibotenic acid as lead compound for the development of novel HDAC7 inhibitors, due to its antiproliferative activity in human breast cancer cells (MCF-7). These promising results represent the starting point for the discovery and the optimization of new HDAC7 inhibitors and highlight the interesting opportunity to apply the “drug repositioning” paradigm also to natural compounds deriving from mushrooms.

## 1. Introduction

Epigenetic alterations are reversible modifications on DNA or histones that affect gene expression without altering the DNA sequence, with an important role in gene regulation [1,2]. Histone modification is one of the primary cellular mechanism involved in the regulation of gene expression. Histone deacetylases (HDACs) are a group of enzymes capable of catalyzing the N-acetyl group’s removal from acetylated lysine residues in histones, regulating fundamental cellular processes, such as cellular proliferation, differentiation, and development [3,4,5]. So far, in mammalian 18 HDACs have been identified and classified into four classes, according to their sequence homologies to yeast. Class I (HDAC1, 2, 3, and 8), class II (HDAC4, 5, 6, 7, 9, and 10), and class IV HDACs (HDAC 11), typically known as “classical HDACs”, are Zn^2+^-dependent enzymes, while class III HDACs (sirtuins 1–7) require nicotinamide adenine dinucleotide (NAD^+^) for their enzymatic activities [6]. Class II is further classified into group IIa and group IIb, based on their structural similarity. The HDAC7 isoform belongs to the group IIa and possesses a large *N*-terminal extension that forms a binding site for several transcription factors and is associated with cellular differentiation. It is expressed in endothelial cells and thymocytes and shuttles from the nucleus to the cytoplasm in response to various regulatory signals [7]. The specific mechanism whereby HDAC7 exerts the transcription repression is not entirely understood. According to the literature reports, probably the *C*-terminal domain of HDAC7 is responsible for its repressive functions [8,9].

In the last decades, increasing evidence has associated epigenetic dysregulation with the development and progression of many diseases, including cancer [10,11,12]. For instance, transcriptional upregulation of HDACs was observed in colorectal, pancreatic, lung, gastric, prostate, and ovarian cancer [13,14,15]. In particular, HDAC 7 was overexpressed in breast cancer [16]. Therefore, HDACs become clinically validated epigenetic drug targets for cancer treatment and HDACs inhibitors (HDACis) have been successfully applied to contrast the cancer progression [17]. Specifically, numerous HDACis showed promising anticancer activities, related to their ability to suppress and induce apoptosis in Michigan Cancer Foundation-7 (MCF-7) breast cancer cell lines [18,19,20]. Overall, HDACis decrease cellular proliferation, induce cell death, apoptosis, and differentiation [15,21]. Despite their promising anticancer potential, severe side effects and toxicities, including thrombocytopenia, neutropenia, vomiting, and diarrhea, resulting from a non-selective inhibition towards various HDACs isoforms, are also documented [4,15]. In this context, the development of new class- or isoform-selective HDACis with improved risk-benefit profiles as anticancer agents is an urgent need.

In recent years, the repurposing of natural products is an emerging approach in the drug discovery programs [22,23,24]. Natural occurring compounds have shown promising biological activities and many natural products have been considered for the treatment of complex diseases, including cancer [25,26]. For instance, Trichostatin A (TSA), isolated from a metabolite of *Streptomyces hygroscopicus*, is one of the most interesting HDACis [27]. It may represent an example of drug repurposing applied to natural products, since first it was studied as a fungistatic antibiotic, and later investigated for its ability to inhibit several HDAC isoforms. Furthermore, natural compounds extracted from plants and mushrooms, such as phenolics, terpenoids, cyclotetrapeptides, alkaloids and hydroxamic acids, have also shown remarkable HDACs inhibitory activities [28,29,30].

In parallel to the repurposing paradigm, the application of in silico techniques for identifying new *hit* compounds is a very profitable approach [31], reducing cost and time of research-associated activities [32,33,34,35]. In particular, structure-based virtual screening (SBVS) is a powerful in silico tool for the identification of novel bioactive ligands and is very useful in predicting the best interaction between ligands and molecular targets [36,37,38,39,40,41,42,43,44,45,46].

In line with our research lines and to figure out novel anticancer strategies, an *in-house* chemical database of extracted molecules from both edible and non-edible mushrooms was virtually screened towards the HDAC7 isoform [47]. The SBVS pointed out the ibotenic acid as a promising HDAC7 inhibitor based on its theoretical binding affinity. In vivo studies demonstrated the capability of ibotenic acid to decrease the cellular viability on MCF-breast cancer cells, providing an interesting example of repurposing natural products to fight cancer disease.

## 2. Results

### 2.1. Structure-Based Virtual Screening (SBVS)

In the current study aimed at identifying new natural chemical entities as anticancer agents, we screened an *in-house* chemical database of molecules extracted from 162 different fungal species, containing 1103 compounds. This database, already used in a previous work against 43 macromolecular targets [47], was filtered off the Pan Assays Interference compounds (PAINS), leading to consider 984 compounds globally. The crystallographic structure of HDAC7 with the Protein Data Bank (PDB) code 3C10 [48] was used to perform the SBVS procedure. In particular, Glide Standard Precision (SP) docking simulation was carried out to evaluate the molecular recognition of 984 compounds against the target receptor. To assess the accuracy and reliability of our docking procedure, we first performed Glide SP re-docking calculations of the reference co-crystallized TSA against HDAC7. The best docking pose of TSA showed a Root Mean Square deviation (RMSD) value, calculated on the ligand heavy atoms, equal to 0.88 Å with respect to the co-crystallized structure (Appendix A). The Glide score (G-Score) of the re-docked X-ray compound (−8.41 kcal/mol) was adopted as *cut-off* to filter the mushrooms database, leading to the selection of 6 *hits*. These molecules showed a better theoretical binding affinity than TSA.

Finally, after a careful visual inspection analysis of the best poses, 3 *hits* (**1-3**) were selected (Table 1).

Structurally, the selected *hits* share a peptidic-like scaffold and are characterized by several polar chemical groups (Table 1). Unfortunately, their commercial availability evaluation revealed that only a compound, the racemic mixture **(R,S)-2**, could be purchased and submitted to further biological assays.

### 2.2. Docking Analysis of the Best Hits

The three best *hits’* molecular recognition analysis highlighted their good accommodation into the HDAC7 catalytic binding site, as shown in Figure 1. In particular, we observed several hydrogen bonds (H-bonds), salt bridges, stacking, and cation interactions between the binding pocket residues of HDAC7 and the new proposed inhibitors, thus rationalizing their excellent theoretical affinity.

By analyzing the best pose of *hit*
**1**, we found its cationic imine group involved in two salt bridges and one π-cation interaction with the side chain of Asp626 and Phe738, respectively. Moreover, the hydroxyl group at position 3 of the β-ribofuranosyl moiety established an H-bond with the imidazole ring of His670 (Figure 1a), involved in crucial interactions with the TSA hydroxamate.

The ibotenic acid is a chiral compound occurring in the S and R forms. In our in silico studies, both isomers were taken into consideration to better understand their binding behavior on HDAC7. The results suggested that both the enantiomers were recognized into the active site, but adopting a different accommodation. In fact, while the amine group of the (S)-ibotenic acid was involved in an H-bond with the His670 residue and in other crucial interactions with the active site residues (Figure 1b), the R-isomer showed a different geometry, thus losing these pivotal interactions (Appendix A). Such a geometrical finding was in line with the binding affinity results, leading to a difference in the G-Score value of almost 2 kcal/mol among the two enantiomers (Appendix A). Based on these observations, we decided to focus our attention only on the *hit*
**(S)-2**, submitted to further computational investigations.

By analyzing the docking pose of *hit*
**(S)-2**, we observed the same interaction of the best docking pose of *hit*
**1**, thanks to its amine group (Figure 1b). Interestingly, the hydroxyl and amine groups of *hit*
**1** and hit **(S)-2**, respectively (Figure 1a,b), seemed to replace and mimic a water molecule, which was found in the active site of the apocdHDAC7 [48].

In addition, we revealed that *hit*
**(S)-2** is able to coordinate the zinc ion by the carboxyl group and the nitrogen atom of the isoxazole ring, as observed for the hydroxyl and the carbonyl oxygens of TSA. Moreover, the carboxyl group is also involved in the same H-bond established between the carbonyl oxygen of the TSA hydroxamide group and a water molecule, which is peculiar to the HDAC7 and absent in the other isoforms. The complex of HDAC7 with *hit*
**(S)-2** was further stabilized by the presence of a π–π interaction between the isoxazole ring of the ligand and the side chain of Hie709, an amino acid residue involved in the coordination of the catalytic zinc and highly conserved in the class I of HDAC.

Finally, *hit*
**3** showed a completely different binding mode with respect to both the X-ray compound and the other selected *hits*. In fact, it was involved in three H-bonds between its urea moiety and 2-pyrrolidone ring with the backbone and the side chain of HDAC7 Gly678 and Asp626, respectively (Figure 1c).

### 2.3. Cell Viability Assay

The correlation between breast cancer and the overexpression of HDAC7 has been widely demonstrated by Witt et al. through proliferation and enzymatic inhibition studies. [16]. Based on a search of the literature, and thanks to data that emerged from computational analyses, in this study MCF-7 cell line was chosen as cellular model of breast cancer to examine hypothetical antineoplastic effects of **(R-S)-2**.

As a matter of fact, the overexpression of HDAC7 in cells MCF-7 cell line determines the up-regulation of many genes with different functions including the pro-metastatic development, the alteration of gene expression and the modulation of associated metabolic pathways [49]. The antineoplastic activity was evaluated through the methylthiazolyldiphenyltetrazolium bromide (MTT) reduction assay, based on the reduction of MTT in the respective formazan by metabolically active cells.

Achieved results are illustrated in Figure 2. Overall, it was valued that the treatment of MCF-7 cells with 10 µM of racemic mixture of the ibotenic acid decreases cell viability approximately by 20–30% in a time-dependent manner, compared to the same untreated cells. This effect is particularly noticeable at 72 h, when racemic mixture of the ibotenic acid causes a significant reduction in cellular activity at least 80%. All these results would encourage further investigations of racemic mixture of the ibotenic acid in the therapeutic oncology field, in order to obtain early preclinical feedback and finally clinical applicability.

### 2.4. Molecular Dynamics Simulation (MDs)

With the aim to study in deep *hit*
**(S)-2** molecular recognition process against HDAC7 and to investigate possible induced-fit phenomena in the receptor binding pocket, the best docking pose of *hit*
**(S)-2** was submitted to explicit water solvent molecular dynamics simulations (MDs). We applied the same protocol to the experimental complex with the reference compound TSA and, after 200 ns of MDs, we observed higher protein stability in the presence of *hit*
**(S)-2** if compared to TSA (Appendix A), as highlighted by the RMSD analysis (RMSD_2_ = 2.45 Å; RMSD_TSA_ = 2.88 Å). Interestingly, after the first 50 ns of the simulation, we observed an increase in the RMSD value of HDAC7 complexed to TSA, due to a more considerable fluctuation of the residues 595–610, as shown in Appendix A. Moreover, the conformational changes, in terms of α-helix, β-strand, and total Secondary Structure Elements (SSE) percentage, were also monitored throughout the whole simulation. Thus, we observed that the total SSE percentage of *hit*
**(S)-2** is slightly higher than TSA, 42.70% and 40.75%, respectively, indicating some differences in the target’s conformational arrangement. In particular, while the percentage change in β-strands was relatively similar between the two complexes (equal to 11.09% and 10.70% for *hit*
**(S)-2** and TSA, respectively), the α-helix composition (equal to 31.62% and 30.05% for *hit*
**(S)-2** and TSA, respectively) showed the most interesting difference. In fact, as shown in Appendix A, the α-helix consisting of residues 595–610 was conserved only in the complex of HDAC7 with *hit*
**(S)-2**, thus justifying its higher geometrical stability if compared to TSA.

During the MDs we also evaluated the geometrical stability of the ligand binding mode, analyzing the RMSD, calculated on the heavy atoms, by initially aligning the complex on the protein backbone of the first MD frame structure (Appendix A). Thus, we observed hit **(S)-2** showed a lower RMSD trend with respect to TSA, indicating a higher geometrical stability of its binding mode. In particular, the RMSF profile of TSA described the dimethylaniline moiety as the most fluctuating portion of the molecule (Appendix A). On the contrary, the idroxamidic acid was found to be stable during the whole simulation due to the good electrostatic interactions established with the residues of the catalytic pocket.

Finally, all the protein interactions with both ligands were monitored throughout the simulation, highlighting that hit **(S)-2** was involved in a more significant number of total contacts than TSA (Figure 3). In particular, Asp707, His709, and Asp801 resulted the most critical residues for binding both ligands to HDAC7 (Figure 3a,b, bottom panel), especially by means of ionic interactions. Similarly, both ligands maintained a stable H-bond with His670 during the whole simulation, while a network of water bridge interactions seemed to improve the proper accommodation of both compounds into the binding site. However, we observed two major differences, since hit **(S)-2** enriched its binding affinity through the formation of two H-bonds with His669 and Gly842, while TSA was able to establish several hydrophobic interactions with Pro542, Phe679, Phe738, and Leu810 due to its aromatic ring and its allylic portion.

## 3. Discussion

Recently, drug repositioning, or drug repurposing, has been proposed as a new and promising direction for drug development [38]. In fact, “the process of finding new uses outside the scope of the original medical indications for existing drugs or compounds” is becoming an attractive strategy thanks to its several advantages, such as significantly cheaper overall development costs and potentially faster progress timelines [50]. Several drugs have been successfully repurposed in past decades, such as thalidomide and sildenafil [51]. Nowadays, due to the new SARS-COV-2 infection, this strategy could offer an immediate and realistic approach to tackle this growing pandemic [52,53].

Moreover, in the last decades, research has increasingly focused on natural compounds with the aim to identify new bioactive compounds to fight several diseases, including cancer. Accordingly, in this work, we considered the mushrooms as an important source of privileged chemical entities for repurposing strategies. According to Chang, the mushrooms can boost the body’s defense mechanism [53] and are endowed with antioxidant, anticancer, immunomodulatory, antimicrobial, anti-inflammatory, and hepatoprotective properties. Specifically, multiple health benefits have been recognized for some mushroom species (*Ganoderma* spp., *Inonotus* spp., *Agaricus* spp. etc.), which exhibit potent medicinal and functional properties [54]. However, the potential of the non-edible mushrooms is totally underestimated. In this context, our experiment allowed us to select three mushroom derivatives as potential HDAC7 inhibitors. In particular, *hit*
**1**, known with the common name of clitidine, is a pyridine nucleoside extracted from the fruiting body of *Clitocybe Acromelalga*, which is a non-edible mushroom. *Hit*
**3** (pyroglutamylcitrulline) is structurally characterized by a pyroglutamil with the citrulline, an α-amino acid isolated from the edible *Agaricus Campestris*. Different studies reported that this mushroom and its extracts showed good antioxidant, antimicrobial, and anticancer activity [55,56,57]. In light of our in silico studies, these two *hits* and their related mushroom species should be further investigated for their potential anticancer profile.

Finally, the ibotenic acid racemic mixture was the only commercially available compound for in vitro testing against the MCF-7 cellular line. It is known that ibotenic acid, extracted from *Amanita* spp., is an excitatory amino acid acting on glutamate, *N*-methyl-d-aspartate (NMDA) and metabotropic receptors, and to a lesser extent with other excitatory amino acids (EAA) receptors [58]. Moreover, it acts as a GABA (gamma-aminobutyric acid) receptor agonist and induces stupor followed by frenzy, affecting the central nervous system [59]. For these reasons, the compound has been used in multiple studies as lead compound for the development of selective ligands for EAA receptors, e.g., 2-amino-2-(3-hydroxy-5-methyl-4-isoxazolyl)acetic acid (AMAA), AMPA and 2-amino-3-(4-bromo-3-hydroxy-5-isoxazolyl)propionic acid (bromo-homoibotenic acid; Br-HIBO), with the former showing potent and selective agonist activity towards NMDA receptors and the other two potently interacting with AMPA receptors [60]. The compound is known for its neurotoxic properties at concentrations ≥25 µM, therefore it is used as a brain-lesioning agent in several experiments to better understand the complex pathologic mechanisms or the pharmacology of the nervous system. Our study highlighted a good anti-proliferative activity of the ibotenic acid at a concentration of 10 µM, thus minimizing or completely abolishing its neurotoxicity [61]. The in vitro assays were performed on MCF-breast cancer cells, a cellular line with overexpression of HDAC7. Therefore, we assumed that the antiproliferative activity of the ibotenic acid could be due to its effect on HDAC7.

The toxicity of this compound could cause some discomfort and risk to cancer patients. Thus, to overcome such adverse effects, the targeted drug delivery could be the most promising approach. In fact, in several clinical studies, novel drug delivery systems, such as liposomes, niosomes, bioadhesive and transdermal systems and others, are associated with a reduced drug toxicity. In addition, the release at the target sites in a controlled manner, may enhance their therapeutic efficiency, improving the bioavailability and, meanwhile, decreasing the side effects [62,63].

## 4. Materials and Methods

### 4.1. Database Preparation

In a previous work [47], we built a database of 1103 natural compounds extracted from 162 fungal species. The identified fungal extracts were collected in the 2D chemical structures from the access websites PubChem [64], ChEMBL [65] and ChemSpider [66]. MarvinSketch [67] was used to draw the missing chemical structures of the above compounds. The protonation state at physiological pH 7.4 was calculated with the LigPrep platform implemented in Schrödinger Suite ver. 11 [68] and the structure of each ligand was optimized using OPLS_2005 as force field [69]. To exclude the possibility of false positive *hits*, the 119 PAINS were deleted from the database by means of the ZINC15 algorithm [70]. Thus, we obtained a final database of 984 *hits* that was submitted to docking simulations.

### 4.2. Receptor Preparation

The 3D coordinates of HDAC7 crystallographic structure were downloaded from the Protein Data Bank website [71] with the PDB code 3C10 [48]. The structure was selected among the several available ones using the following criteria: (a) best crystallographic resolution; (b) source from *Homo sapiens*; (c) the presence of one co-crystallized ligand; (d) the PDB deposition data.

The PDB was prepared in order to assign the correct bond orders, to build missing atoms, side chains, and loops and to add all the hydrogen atoms as implemented in the Protein Preparation Wizard tool [72,73,74]. Co-crystallized water molecules were removed, while we preserved a single water molecule near the active site involved in an H-bond with Gly842 during the catalysis [48].

### 4.3. Docking Simulations

The optimized HDAC7 structure was used to perform docking calculations by adopting the software Glide ver. 7.2 [75]. The co-crystallized ligand was used to center the rigid receptor grid, defined by a 10 × 10 × 10 Å inner box. The Glide SP protocol was applied with the default parameters, generating 10 poses per ligand.

A preliminary validation of the computational docking protocol was performed, adopting the same parameters, with the aim to evaluate the capability of the Glide SP to geometrically reproduce the crystallographic pose of the reference compound TSA against the HDAC7 model.

Thus, the mushroom database was screened towards the HDAC7 receptor. In order to select the most promising *hits*, as a *cut-off* we used a G-Score value equal to −8.41 kcal/mol, obtained from the analysis of the best TSA re-docking pose.

### 4.4. Molecular Dynamics Simulations (MDs)

The complexes of HDAC7 with the ibotenic acid and TSA were submitted to 200 *ns* of MD simulations using Desmond ver. 4.4 [76]. Both systems were placed in a 10 Å layer orthorhombic box in explicit solvent with TIP3P [77] water model parameters. In total, 5 K^+^ counterions were added to the systems until charge neutralization. After optimizing the solvated model, both systems were relaxed with the Martyna-Tobias_Klein isobaric-isothermal ensemble (MTK_NPT) and then equilibrated through the NVT ensemble at 10 K, by using the NPT ensemble at 300 K and 1 atm with the Berendsen thermostat-barostat. Trajectory frames were collected every 50 *ps* and analyzed by means of the Simulation Interaction Diagram and the Simulation Event Analysis, in order to geometrically and thermodynamically investigate the obtained trajectories.

### 4.5. Cell Viability Assay

The MTT assay was performed on MCF-7 (a human breast cancer cells), grown in DMEM medium (Sigma Aldrich, St. Louis, MO, USA) supplemented with 10% fetal bovine serum (FBS) (Sigma Aldrich, St. Louis, MO, USA), 1% Penicillin/Streptomycin (Sigma Aldrich, St. Louis, MO, USA) at 37 °C in a 5% CO2 incubator. MCF-7 cells were exposed to racemic mixture of ibotenic acid. Stock solution of **(R,S)**-was dissolved in DMSO. When cells were confluent, 10^4^ cells were plated in a 96-well and treated with **(R,S)-2** (10 μM) two hours later. DMSO was used ad control. MTT was carried out according to the protocol of Costa G. et al. [36].

## 5. Conclusions

In this work, in silico studies and biological assays were performed with the aim to explore the potential anticancer activity of natural compounds extracted from mushroom species. Specifically, our SBVS protocol pointed out clitidine, ibotenic acid, and pyroglutamylcitrulline as promising HDAC7 inhibitors. In particular, MTT assay on MCF cells demonstrated that the ibotenic acid at 10 µM was able to decrease the cellular viability in a time-dependent manner. As far as our knowledge goes, no neurotoxic effects are reported for ibotenic acid at the concentration used in the current study. Therefore, the ibotenic acid could represent an example of natural drug repurposing approach and the bioactive compounds present in the mushrooms could be considered as an important source of not yet deeply explored chemical entities, especially for the underestimated non-edible fungi.

## Figures and Tables

**Figure 1 molecules-25-05524-f001:**
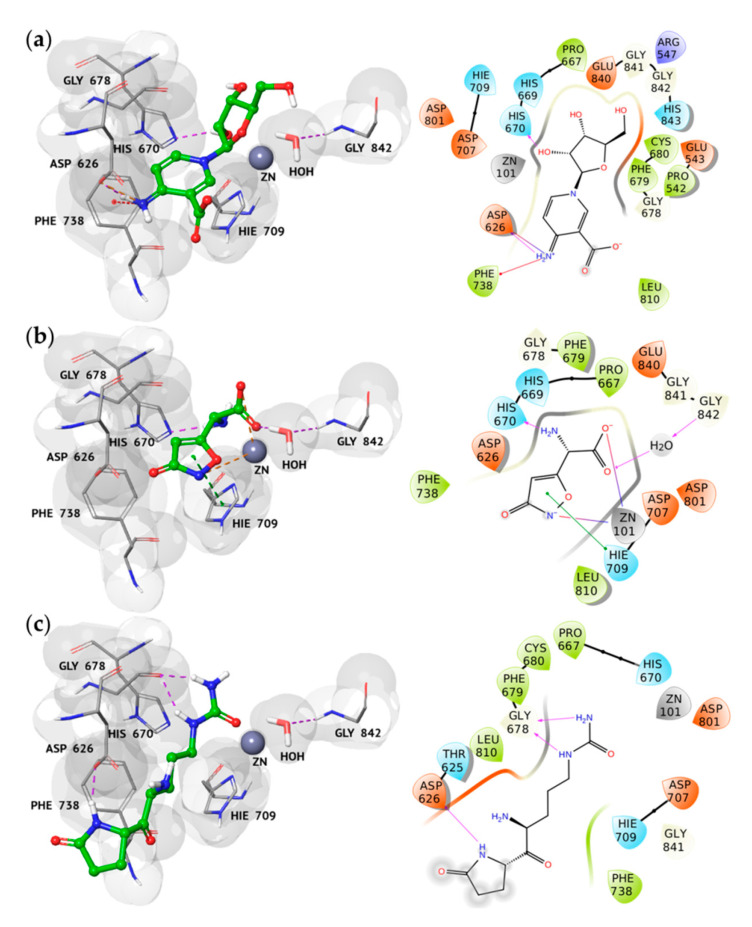
3D and 2D representations of the best *hits*
**1** (**a**), **(S)-2** (**b**), and **3** (**c**) complexed to HDAC7 (PDB code 3C10), respectively. Protein is shown as grey surface, ligands are displayed as green carbon ball-and-sticks, while the amino acid residues involved in the molecular interactions are reported as grey carbon sticks. In the 2D representations, H-bonds, salt bridges, stacking, and π-cation interactions are depicted as violet, orange, green, and red lines, respectively.

**Figure 2 molecules-25-05524-f002:**
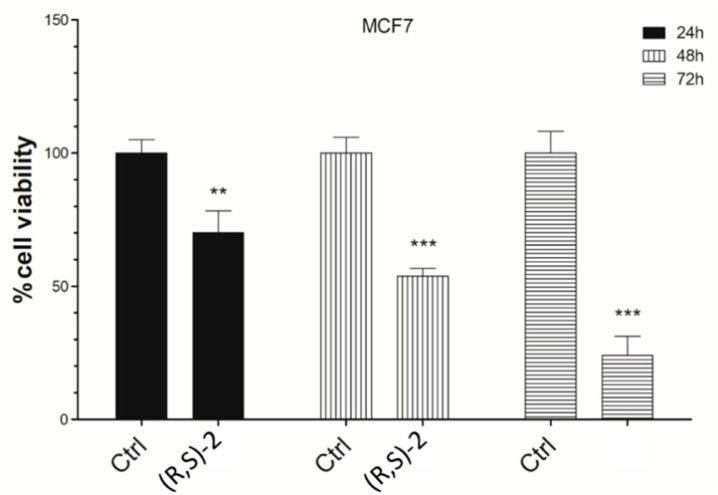
Effects of racemic mixture **(R,S)-2** on MCF-7 cell viability. In total, 10^4^ cells were seeded in triplicate in 96-well plates and treated with **(R,S)-2** at concentration of 10 μM, cells treated with DMSO were used as control. Cell viability was measured by an MTT assay at 24, 48, and 72 h after treatment. Results are expressed as a percentage of control, analyzed by ANOVA (** *p* < 0.005; *** *p* < 0.0005), each column represents the mean ± SD of three different wells.

**Figure 3 molecules-25-05524-f003:**
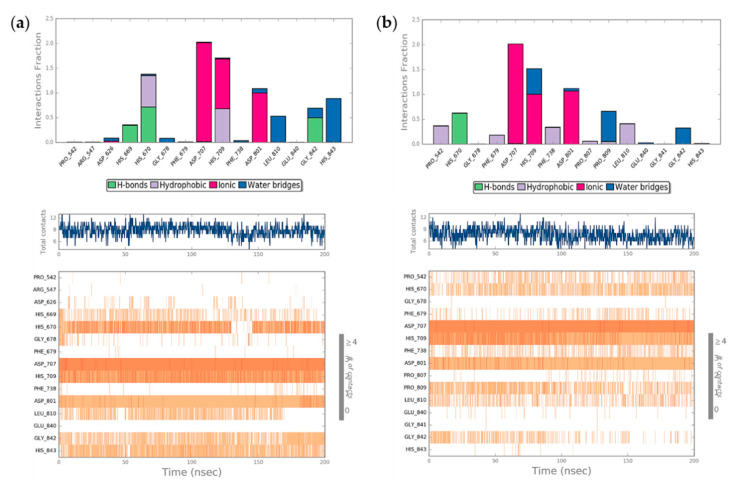
Protein interaction analysis of hit **(S)-2** (**a**) and TSA (**b**), monitored throughout the MD simulation. In particular, the fraction of ligand-protein interaction time for protein residues during the simulation is represented in the top panel, while in the bottom the timeline representation of the interactions and contacts throughout the MD simulation is reported.

**Table 1 molecules-25-05524-t001:** *Hit*, Name, 2D structure, and G-Score value (kcal/mol) of the 3 best *hits* proposed as potential HDAC7 inhibitors and TSA.

*Hit*	Name	2D Structure	G-Score (kcal/mol)
**1**	Clitidine	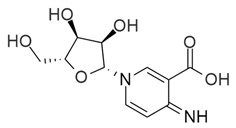	−8.73
**(S)-2**	(S)-Ibotenic acid	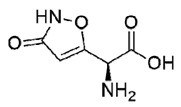	−8.60
**3**	Pyroglutamylcitrulline	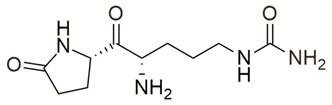	−8.91
**4**	Trichostatin A	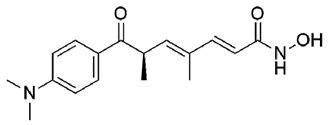	−8.41

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
