# Peer review of "Natural Products Extracted from Fungal Species as New Potential Anti-Cancer Drugs: A Structure-Based Drug Repurposing Approach Targeting HDAC7"

_molecules, 2020, doi:10.3390/molecules25235524_

Round 1

Reviewer 1 Report

In this manuscript, by using virtual screening, Maruca and co-workers set out to identify potential HDAC7 inhibitors from a virtual library of compounds extracted from edible and non-edible mushrooms. Three compounds were identified from the virtual screen, and one of them (ibotenic acid) was tested against MCF7 cells using the MTT cell viability assay. MD simulations and docking studies were also performed to predict the binding mode(s) of the compound.

Major issue:

This is a very preliminary study. The only experimental testing was a cell viability assay performed with MCF7 cells. There is no direct evidence that ibotenic acid actually binds to HDAC7. There is also no direct evidence that show ibotenic acid is an inhibitor of HDAC7. Without these two key pieces of information, the computational work that was performed in this study has no value on its own. Based on these reasons, I cannot recommend publication of this manuscript in its current form.

In order to get this paper published, the authors should perform the following in vitro experiments with recombinant HDAC7 enzyme (the enzyme is commercially available). First, the authors should perform binding studies to characterise the binding interactions between ibotenic acid and HDAC7 (e.g. Kd determination). Secondly, competition binding experiments with a known HDAC inhibitor (e.g. Trichostatin A, which is commercially available) should be conducted to show that ibotenic acid binds to the active site of the enzyme. Finally, inhibition assay should able be performed to obtain the IC50 value of ibotenic acid and then compare the value with other known HDAC7 inhibitors (e.g. by using commercially available HDAC activity assay kit).

Minor issue:

(1) Structure of Trichostatin A (the ligand in 3C10) and its redocked G-score should be shown in Table 1.

(2) There are minor spelling and grammatical errors throughout the manuscript

Author Response

Answers to Reviewer 1                                                             

In this manuscript, by using virtual screening, Maruca and co-workers set out to identify potential HDAC7 inhibitors from a virtual library of compounds extracted from edible and non-edible mushrooms. Three compounds were identified from the virtual screen, and one of them (ibotenic acid) was tested against MCF7 cells using the MTT cell viability assay. MD simulations and docking studies were also performed to predict the binding mode(s) of the compound.

Major point.

This is a very preliminary study. The only experimental testing was a cell viability assay performed with MCF7 cells. There is no direct evidence that ibotenic acid actually binds to HDAC7. There is also no direct evidence that show ibotenic acid is an inhibitor of HDAC7. Without these two key pieces of information, the computational work that was performed in this study has no value on its own. Based on these reasons, I cannot recommend publication of this manuscript in its current form.

In order to get this paper published, the authors should perform the following in vitro experiments with recombinant HDAC7 enzyme (the enzyme is commercially available). First, the authors should perform binding studies to characterise the binding interactions between ibotenic acid and HDAC7 (e.g. Kd determination). Secondly, competition binding experiments with a known HDAC inhibitor (e.g. Trichostatin A, which is commercially available) should be conducted to show that ibotenic acid binds to the active site of the enzyme. Finally, inhibition assay should able be performed to obtain the IC50 value of ibotenic acid and then compare the value with other known HDAC7 inhibitors (e.g. by using commercially available HDAC activity assay kit).

We are grateful to the reviewer for his observations. Unfortunately, we have no opportunities to validate the interaction between HDAC7 and ibotenic acid due to the difficulties that we are experiencing in our lab due to the new Covid-19 lockdown.

However, in our manuscript we supply for the first time the proof of principle that ibotenic acid, as a bona fide HDAC7 inhibitor, has anticancer activity. Our in vitro experiments have been carried out on MCF-7, an overexpressing HDAC7 human mammary cancer cell line, as described in the literature. We sincerely believe that this finding is interesting enough to be published in Molecules and, should we be able to leave behind the pandemic restrictions, it is our intention to pursue the suggestions we received by this reviewer in the next future.

The text of the manuscript was modified accordingly in both Results and Methods.

Minor issue:

(1) Structure of Trichostatin A (the ligand in 3C10) and its redocked G-score should be shown in Table 1.

We added both the structure and the G-score of Trichostatin A in Table 1.

(2) There are minor spelling and grammatical errors throughout the manuscript.

We thank the reviewer for all suggestion, we checked and corrected the spelling and grammatical errors throughout the manuscript.

Reviewer 2 Report

This manuscript it is an interesting approach to target HDAC7. The authors did silico studies and biological assays with the aim to explore the potential anticancer activity of natural compounds extracted from mushroom species, targeting HDAC7 receptor. Specifically, their protocol pointed out clitidine, ibotenic acid, and pyroglutamylcitrulline as promising HDAC7 inhibitors. In particular, MTT assay on MCF cells demonstrated that the ibotenic acid at 10 μM was able to decrease the cellular viability in a time dependent manner. Therefore, the ibotenic acid could represent an example of natural drug repurposing approach.

Only for this result this manuscript should be published in molecules as will be very useful for many researchers.

I include very minor revisions to be done
1. add space between [54] and However in line 210

2. The year should be bold in reference 32.

Author Response

Answers to Reviewer 2

This manuscript it is an interesting approach to target HDAC7. The authors did silico studies and biological assays with the aim to explore the potential anticancer activity of natural compounds extracted from mushroom species, targeting HDAC7 receptor. Specifically, their protocol pointed out clitidine, ibotenic acid, and pyroglutamylcitrulline as promising HDAC7 inhibitors. In particular, MTT assay on MCF cells demonstrated that the ibotenic acid at 10 μM was able to decrease the cellular viability in a time dependent manner. Therefore, the ibotenic acid could represent an example of natural drug repurposing approach.

Only for this result this manuscript should be published in molecules as will be very useful for many researchers.

We thank very much the reviewer for her/his comment.

Minor point.

Add space between [54] and However in line 210

At line 210 we added the requested space.

The year should be bold in reference 32.

 The year of reference 32 is now in bold.

Reviewer 3 Report

Reviewer reports for Maruca et al. “Natural products extracted from fungal species as new potential anti-cancer drugs: a structure based drug repurposing approach targeting HDAC7”

Molecules 994142

This work describes an application of the drug-repurposing approach to natural products isolated from mushrooms. Three secondary metabolites identified from an in-house database were investigated as inhibitors of histone deacetylases, targets that are known to be involved in different cancer processes. From an in-silico interaction analysis and molecular dynamics studies clitidine, ibotenic acid and pyroglutamylcitrulline were identified as promising HDAC-7 inhibitors, and ibotenic acid showed cell viability at 10 micromolar concentration.

Overall, I think that this is a carefully conducted and interesting study on a timely subject. I recommend publication, but suggest a few minor revisions:

  1. a) in table 1 the structure of ibotenic acid is shown in racemic form. In figure 1 (interaction studies) the naturally occurring L-form is shown. Commercially available is racemic ibotenic acid. What did the authors use for the cell viability study? If it was racemic ibotenic acid they should state this explicitly in the discussion, and they should also discuss how the use of the racemate will most likely affect binding affinities. It would be interesting to see separate binding studies for both isomers of ibotenic acid – if this is doable, I would urge the authors to include this interesting investigation in this paper.
  2. b) a few minor typos:

- line 50: evidence, not evidences

- line 64: remove two out of three full stops before and after [22]

- line 213: which is a non-edible mushroom

- line 216: reported that this mushroom...

- lines 488 ff: provide last date of access for internet sources. Check if citing these internet sources can be avoided by citing alternative references.

Author Response

Answers to Reviewer 3

This work describes an application of the drug-repurposing approach to natural products isolated from mushrooms. Three secondary metabolites identified from an in-house database were investigated as inhibitors of histone deacetylases, targets that are known to be involved in different cancer processes. From an in-silico interaction analysis and molecular dynamics studies clitidine, ibotenic acid and pyroglutamylcitrulline were identified as promising HDAC-7 inhibitors, and ibotenic acid showed cell viability at 10 micromolar concentration.

Overall, I think that this is a carefully conducted and interesting study on a timely subject. I recommend publication, but suggest a few minor revisions.

We thank very much the reviewer for her/his comment.

Minor point.

In table 1 the structure of ibotenic acid is shown in racemic form. In figure 1 (interaction studies) the naturally occurring L-form is shown. Commercially available is racemic ibotenic acid. What did the authors use for the cell viability study? If it was racemic ibotenic acid they should state this explicitly in the discussion, and they should also discuss how the use of the racemate will most likely affect binding affinities. It would be interesting to see separate binding studies for both isomers of ibotenic acid – if this is doable, I would urge the authors to include this interesting investigation in this paper.

According to the reviewer suggestion, we clarified the difference of the enantiomers of the ibotenic acid and we implemented the related in silico investigation on both isomers. We added both 2D and 3D representations of the (R)-Ibotenic acid in the Supplementary Material, while, regarding the biological assay, for hit 2 we specified the racemic mixture (R,S)-2.

line 50: evidence, not evidences

line 64: remove two out of three full stops before and after [22]

line 213: which is a non-edible mushroom

line 216: reported that this mushroom...

All minor revisions have been made.

- lines 488 ff: provide last date of access for internet sources. Check if citing these internet sources can be avoided by citing alternative references.

As suggested, we provided the last date of access for internet sources.

Round 2

Reviewer 1 Report

I have gone through the revised manuscript and the author’s reply to my comments to the original manuscript. The authors’ argument is that “HDAC7 is overexpressed in MCF-7 cell lines, and that ibotenic acid appears to have antiproliferative activity against MCF-7 means that it is possible that ibotenic acid is targeting HDAC7” only shows half the story. If we are going to follow the same line of argument, then the authors need to conduct control experiments with cell lines that are not overexpressed with MCF-7. Even then, it is difficult to directly pinpoint the effect onto MCF-7 inhibition. Personally, I feel that it is necessary to conduct the in vitro binding / inhibition experiments with purified HDAC7 to justify the claims that the authors are making in the manuscript. I think that, in order to have a paper that is actually going to make an impact to the community, these controls need to be conducted

However, I also understand the impact of COVID-19 especially in Italy where the authors are based - so I will leave that to the editor to make the decision. In case the editor would like to accept the manuscript (I can understand although I cannot enthusiastically agree), I strongly recommend the authors to rewrite the discussions section, and to clearly highlight the caveats of the current studies, e.g. “The in vitro studies were conducted with MCF-7, a cell line with overexpression of HDAC7. It is possible that the observed antiproliferation effect could be due to HDAC7 inhibition or other off-target effect. Future experiments, including XXXXXX, are needed to confirm whether ibotenic acid is actually targeting HDAC7.”